# Neural-Network-Based Localization Method for Wi-Fi Fingerprint Indoor Localization

**DOI:** 10.3390/s23156992

**Published:** 2023-08-07

**Authors:** Hui Zhu, Li Cheng, Xuan Li, Haiwen Yuan

**Affiliations:** College of Electrical Information, Wuhan Institute of Technology, Wuhan 430205, China

**Keywords:** convolutional neural network, 3D ray tracing, radio signal strength, Wi-Fi indoor localization

## Abstract

Despite the high demand for Internet location service applications, Wi-Fi indoor localization often suffers from time- and labor-intensive data collection processes. This study proposes a novel indoor localization model that utilizes fingerprinting technology based on a convolutional neural network to address this issue. The aim is to enhance Wi-Fi indoor localization by streamlining the data collection process. The proposed indoor localization model leverages a 3D ray-tracing technique to simulate the wireless received signal strength intensity (RSSI) across the field. By incorporating this advanced technique, the model aims to improve the accuracy and efficiency of Wi-Fi indoor localization. In addition, an RSSI heatmap fingerprint dataset generated from the ray-tracing simulation is trained on the proposed indoor localization model. To optimize and evaluate the model’s performance in real-world scenarios, experiments were conducted using simulated datasets obtained from the publicly available databases of UJIIndoorLoc and Wireless InSite. The results show that the new approach solves the problem of resource limitation while achieving a verification accuracy of up to 99.09%.

## 1. Introduction

With the application and development of related technologies based on user location information, location-based services have become essential for daily work and life [1]. This is particularly crucial in large and complex indoor environments, such as museums, airports, supermarkets, hospitals, underground mines, and other areas in which there is an urgent need for location-based services. Positioning technologies can be classified into two types: outdoor and indoor positioning technologies. In an outdoor environment, global-positioning systems, BeiDou-positioning systems, and other global navigation satellite systems (GNSSs) can provide users with meter-level location services widely utilized in daily activities, providing accurate positioning in outdoor spaces [2]. The global-positioning system (GPS) has made navigation systems practical for many land vehicle applications. Abbott et al. [3] introduced a method of integrating a GPS with a simplified inertial navigation system (INS) and provided a technique of using velocity aiding to improve positioning accuracy and reliability. With the increasing popularity of mobile devices with positioning capabilities, such as GPS phones, Zheng et al. [4] applied a collective matrix factorization method to mine interesting locations and activities. They used them to recommend to users areas for performing specific exercises and activities to participate in when visiting particular sites. GPS-based location services provide more convenient and effective technical support for outdoor positioning.

However, in an indoor environment, where humans are located 80% of the time, GNSS positioning accuracy is drastically reduced owing to the obscuration of buildings and the multipath effect, hindering the satisfaction of the demand for accurate indoor location services [5]. With the development of wireless indoor positioning technologies, such as Wi-Fi, Bluetooth, and ultra-wideband technology, various indoor positioning technologies and systems have been proposed for providing location services in large buildings [6]. Bluetooth and Wi-Fi indoor localization are two standard wireless signal-based localization techniques. They can both utilize the features of wireless signals, such as received signal strength indicators (RSSIs) or channel state information (CSI), to estimate location information. They can also adopt the fingerprinting method, which builds a fingerprint database by collecting the signal features at different locations in advance and determines the optimal location by matching algorithms.

Indoor localization has many application scenarios and practical needs, such as emergency management, navigation services, logistics management, smart homes, etc. Filippoupolitis et al. [7] proposed to use of Bluetooth low-energy (BLE) technology to address the occupancy problem in emergency management using beacons installed in buildings to provide the location information of users and combining machine-learning methods to determine whether there were occupants in specific areas. Moreover, in intelligent energy management, Tekler et al. [8] proposed a novel plug load management system that also combined BLE and machine-learning methods to determine occupancy in specific areas and reduced the plug load energy consumption and user burden through intelligent plug load automation. Balaji et al. [9] proposed leveraging existing Wi-Fi infrastructure in commercial buildings and smartphones carried by building occupants to provide occupancy-based fine-grained HVAC actuation in a smart home domain. Tekler et al. [10] used a feature selection algorithm to select the most essential features from sensor data. Then, they used different deep-learning models to predict occupancy based on these features.

Owing to the large availability of existing infrastructure, Wi-Fi is widely used in homes, hotels, cafes, airports, shopping malls, and other large or small buildings, making Wi-Fi one of the most compelling wireless technologies for location services [11]. Considering the ubiquity of mobile devices and routers in the experimental site and the comprehensive coverage of Wi-Fi signals, this paper chooses to use Wi-Fi technology for this research. Typically, a Wi-Fi system consists of several fixed access points (APs) deployed in locations known by the system or network administrator that provide easy accessibility and installation. Mobile devices that can connect to Wi-Fi (e.g., laptops and cell phones) can communicate with each other directly or indirectly (through APs), permitting the implementation of a location function in addition to a communication function [12]. This Wi-Fi positioning system with fingerprinting technology is becoming increasingly popular, and using the ubiquitous received signal strength intensity (RSSI) signal received by a Wi-Fi device for positioning is an effective way to identify a user’s location in indoor environments. To measure the distance between nodes, the RSSI (received signal strength indicator)-ranging technique utilizes the principle of regular signal attenuation with increasing distance for wireless signals [13]. The signal strength of a transmitting node can be obtained from an RF chip register. Based on the received signal strength, the receiving node calculates the transmission loss of the signal and converts it to distance using a theoretical or empirical model [14]. This ranging technique only requires a wireless transceiver at a node; no additional hardware is needed, keeping the application cost low.

With the advent of artificial intelligence, the challenges faced in indoor positioning have provoked the use of deep learning to improve the efficiency of positioning frameworks, ushering in cross-era changes. Chen et al. [15] proposed multisource information fusion positioning technology to effectively utilize Wi-Fi fingerprint data and the geomagnetic field for positioning, addressing the problem that Wi-Fi signals are unstable in complex indoor environments and buildings distort the local geomagnetic field, resulting in low positioning accuracy at a single location source. Liu et al. [16] proposed a joint convolutional neural network (CNN)-based channel state information (CSI) fingerprint indoor localization method to obtain average positioning errors of 24.7 cm and 48.1 cm in two positioning scenarios in a gallery and a laboratory, respectively, and verified that the joint localization algorithm was effective.

Considering the advantages of Wi-Fi fingerprint-based indoor localization methods combined with neural network methods, this study proposes a CNN-based fingerprint indoor localization model consisting of two stages. In the offline stage, a fingerprint database containing all the reference points in the localization area is constructed for offline training. Meanwhile, in the online scene, an algorithm is applied to match real-time fingerprint information from a user with the offline fingerprint database and estimate the user’s location. This algorithm addresses the problem of the limitation of resources in front-end data collection while providing a low-cost and high-accuracy indoor positioning solution. The main contributions of this study are as follows:

1. 3D ray-tracing technology is proposed to generate RSSI signals in the localization area, as simulated location information can avoid the inherent noise and instability of actual wireless signals, which can cause instability in localization performance.

2. To address small-scale intensive localization needs and tackle the problem of minor differences in RSSI signal characteristics among APs in localization, the construction of a Wi-Fi fingerprint heatmap set is proposed, which can better characterize the differences in intensity characteristics at different reception points.

3. The lack of localization accuracy demonstrated by traditional CNN models is improved upon in this study, providing a framework with excellent localization performances for areas with different depths.

4. Experiments are conducted with the synthetically created and UJIIndoorLoc indoor localization datasets [17]. The simulated and actual measurement results verify the effectiveness of our proposed localization method.

## 2. Related Work

An indoor positioning system effectively uses Wi-Fi APs and radio signal strength (RSS) to facilitate localization [18]. However, implementing a fingerprint-based approach requires time-consuming radio surveys and data acquisition to construct a database for each building. The task of front-end data collection is costly in both time and labor. RSS values are incredibly dependent on the environment, making front-end data collection exceptionally difficult, resulting in meager qualification rates of data collected in the field, which do not meet current survey standards [19]. In some large-scale scenarios of localization, Zhang et al. [20] proposed to convert collected Bluetooth RSS into fingerprint images required for calculation and establish a CNN for classification training. However, tedious collection work is often needed before research is carried out. In addition, different devices have different signal sensitivities, and data elimination is an important step. Liu et al. [21] proposed constructing a ratio fingerprint by calculating the ratios of different RSSIs from important contribution access points, which somewhat alleviated the collection work. However, in small-scale scenarios, the percentages of different RSSIs from important contribution access points were also reduced, and the method of constructing ratio fingerprints was no longer suitable for this scenario.

To address the above issues, Li et al. [22] uploaded the RSSI fluctuations of detected Bluetooth nodes to the cloud and performed real-time correction of the RSSI values. Sinha et al. [23] simulated constantly varying RSSI values based on reference RSSI values to achieve data augmentation. Both methods processed the data at the front end, significantly saving resources in front-end data collection. Sun et al. [24] verified that the deployment of radio mapping could dramatically reduce the front-end data collection effort. However, the inherent fluctuation in RSSs generally does not guarantee that the position containing the highest probability predicted by each classifier is actual, resulting in a severe barrier to desired performance in existing fusion methods. To overcome these drawbacks, Hashem et al. [25] proposed the design and implementation of WiNar, a Wi-Fi indoor location determination system based on the round-trip time that combines the advantages of fingerprinting and range-based techniques to overcome the various challenges of indoor environments. A localization model based on CNNs and extended short-term memory networks was also proposed [26]. Guo et al. [27] used the k-nearest neighbors (KNN) algorithm and outlier detection methods to construct an indoor localization framework for simple fingerprints. The existence of just a single evaluation index and poor adaptability to outlier detection hindered the ability of this framework to achieve significant improvement in localization performance. Xie et al. [28] proposed using a back-propagation (BP) neural network and a weighted KNN algorithm to obtain higher localization accuracy. Wi-Fi indoor localization is highly environment-dependent; however, the BP algorithm was susceptible to initial weights.

These studies highlight the difficulties in providing accurate indoor localization, described as follows. Due to a lack of front-end resources, data collection is labor-intensive and costly under resource constraints. Uneven data cause unsatisfactory localization performances. Finally, improvement in the localization capability of models is hindered by an imbalance in training sample features.

In this study, we use a 3D ray-tracing technique to construct fingerprint datasets of a localization area [29], addressing the problems of data contamination and consumption in field collection. Furthermore, considering that the differences in data features in self-constructed localization areas are typically too small, we propose the construction of a fingerprint heatmap to characterize the uniqueness of sample features.

To validate the performance of the proposed model in localization, experiments are conducted using the indoor positioning database UJIIndoorLoc and the simulation database, Remcom Wireless InSite 3.3.0 [30]. The results show that the proposed model performs well regarding localization accuracy while ensuring low power consumption.

## 3. System Model

The traditional fingerprint-based indoor positioning process is illustrated in Figure 1, mainly separated into offline and online phases. In this process, data are collected, fingerprint datasets are constructed offline, and fingerprint data matching is performed online to obtain a user’s location [31]. This fingerprint-based indoor positioning method often requires high cost and time-consuming labor in the offline phase, and the qualification rate of the collected data is shallow, which does not meet the survey standard at this stage.

We improved the traditional indoor positioning process by considering the differences in positioning data for different positioning scenarios in this study, and we show a depiction of the constructed system model in Figure 2. In the offline phase, data were collected, and fingerprint datasets were built. The original data for large-scale scenarios were used with the UJIIndoorLoc dataset, and the actual data for small-scale scenarios were used with the Remcom Wireless InSite 3.3.0 software. Due to the difference in data density between the large-scale and small-scale scenarios, grayscale and thermal fingerprint maps were obtained for these scenarios after fingerprint processing. A fingerprint dataset was also constructed and input into the localization network for training to receive the model’s weights. In the online phase, the raw fingerprint data of the collected unknown points were transformed into corresponding fingerprint maps according to the scenario’s requirements, and the localization information of the desired points was retrieved from the model weights in the offline phase.

### 3.1. RSSI Loss Modeling

The fingerprint-based localization method used the wireless signal strength fingerprint to achieve localization. Wireless signal strength can be expressed as a function of distance and the path loss exponent, which captures the effect of the environment on the signal attenuation, as shown in Equation (1). In the equation, $P_{t}$ is the transmit power, n is the path loss exponent, d is the distance from the radio source, and $d_{0}$ is a reference distance [32]. This equation implies that the wireless signal strength decreases logarithmically with length increase.
(1)$$RSSI\left(dBm\right)=P_{t}\left(dBm\right)-10n\cdot {log}_{10}\frac{d}{d_{0}}$$

The received power $P_{r}$ is inversely proportional to the square of the free-space distance d between the transmitter and the receiver, which is given by Equation (2):(2)$$P_{r}=P_{t}\cdot G_{t}\cdot G_{r}{\left(\frac{\lambda }{4\pi d}\right)}^{2}$$
where $P_{t}$ is the transmission power, $G_{t}$ is the transmitter gain, $G_{r}\;$ is the receiver gain, and λ is the wavelength [33]. Researchers often use the inverse relationship between received power and the distance to a transmitter at a known location, usually measured in meters or kilometers, to locate wireless transmitter receivers [34].

### 3.2. Convolutional Neural Network Mode

The typical structure of a CNN is shown in Figure 3. A CNN consists of an input layer (input is a digital matrix of the original image), a stacked convolutional layer, a pooling layer, a fully connected layer, and an output layer [35].

#### 3.2.1. Convolutional Layer

Convolution in a convolutional layer refers to calculating an inner product based on multiple convolution kernels with certain weights to perform internal product operations on a local set of pixels from an input image or feature. The final output value obtained is one of the extracted features.

#### 3.2.2. Pooling Layer

The pooling layer is a sampling layer that downsamples the output features of the previous convolutional layer. It further reduces the algorithm’s computational complexity by filtering features and reducing the size of the feature matrix. The main advantages of a pooling layer are feature dimensionality reduction, reduction in overfitting, and improvement in the fault tolerance of a model.

#### 3.2.3. Fully Connected Layer

After convolution, excitation, and pooling, the data are input to the fully connected layer. The main reason for using a fully connected layer is that too many neurons before the fully connected layer cause a network to learn too much, resulting in overfitting. Therefore, it is necessary to introduce a dropout operation to perform a routine to increase the robustness of a model, such as randomly removing some neurons in the neural network or performing local normalization and data augmentation. After the data reach the fully connected layer, the network can be treated as a simple multiclassification neural network, and the SoftMax function can obtain the final data.

## 4. Indoor Position Methods

In response to the lack of front-end data in existing Wi-Fi indoor localization methods, this study proposes a Wi-Fi indoor localization method based on an improved CNN. The indoor localization flowchart presented in this study is shown in Figure 4 and Figure 5. The criterion for distinguishing between large-scale and small-scale scenarios was whether room-level or building- and floor-level localization were performed. Figure 4 shows a large-scale technique for building- and floor-level localization. In the first step, the RSSI values of the reference points of the large-scale locations were collected. The collected data were processed in the second step to construct fingerprint grayscale datasets. In the third step, training was performed on the improved CNN. Finally, the localization results were obtained. Figure 5 shows a small-scale scenario for room-level localization. For small-scale locations, an environment was constructed and simulated to obtain the RSSI values of the reference points in the first step. The construction of a thermal fingerprint dataset followed this. Then, training occurred with the improved CNN, followed by the output of the localization results. These two scenarios could be performed separately according to specific needs, or they could be integrated for procedures that required both. Therefore, this could more finely meet the localization needs of different techniques.

### 4.1. Floor-Level Large-Scale Dataset Processing

Computers recognize images simply by identifying the colors of pixels, and image storage also involves the conversion of colors to a numeric type to yield a sizeable numeric matrix by which the image information is stored [36]. As CNNs are generally composed of RGB values for image recognition tasks, this color feature is handy for recognizing objects of different categories. However, within the same type, color is not as important; texture features are more important. As RSSI signal strength attenuation between neighboring APs located in large-scale scenes is readily apparent and it is easier to obtain prominent differences in location features when processing fingerprints at the image level, this paper created a floor-level large-scale dataset by combining a single-channel Wi-Fi fingerprint grayscale map with the UJIIndoorLoc database.

#### UJIIndoorLoc Dataset

In this study, we used the UJIIndoorLoc database to process large-scale data for indoor localization. This publicly available dataset contains WLAN fingerprinting data collected from three buildings of Universitat Jaume I with four or more floors and an area of approximately 110,000 m^2^. The data were gathered from over 20 users and 25 Android devices in 2013. The database has 19,937 training records and 1111 validation records, each with 529 attributes. The attributes include the Wi-Fi fingerprint, the coordinates, the building ID, the floor ID, and other information, such as the user, the device, the timestamp, the space, and the relative position. The Wi-Fi fingerprint consists of 520 intensity values of the received signal strength intensity (RSSI) of detected wireless access points (WAPs). The RSSI values range from −104 dBm (inferior signal) to 0 dBm, and 100 indicates that a WAP was not detected. The database can be used for classification or regression tasks, such as identifying the building and floor or estimating the latitude and longitude of a user. Here, −110 dBm was used to denote a WAP that was not detected. The following describes the method for processing the fingerprint grayscale map to achieve a better performance with the proposed localization scheme.

To obtain the wireless signal strength fingerprint, $RSSI_{m}^{n,k}$, it was denoted as the offline wireless signal strength database. *m* is the index of the location acquisition point for which m=1,2,3,···,M; *M* is the total number of location acquisition points set in the indoor environment; *n* is the index of a fingerprint collected at the nth acquisition point, where n=1,2,3,···,N; *N* is the total number of fingerprint samples collected; *j* denotes the index of the AP where j=1,2,3,···,J; and *J* is the total number of APs available in the environment. For maximum wireless signal strength $RSSI_{Max}$ and minimum wireless signal strength $RSSI_{Min}$, the normalized distribution of the wireless signal strength could be obtained from Equation (3), as follows:(3)$$RSSI_{m,new}^{n,k}=\frac{RSSI_{m}^{n,k}-RSSI_{Min}}{RSSI_{Max}-RSSI_{Min}}$$

By processing the source, it was possible to obtain distributions for many acquired Wi-Fi intensities. As introduced above, each raw fingerprint datapoint in the UJIIndoorLoc dataset consisted of 520 RSSI (received signal strength intensity) values from the received wireless access points. To fully display them on the fingerprint grayscale map, we filled in the missing data with zeros after transforming them into a 24 × 24 matrix to expand each fingerprint datapoint to 1 × 576. Each fingerprint record obtained was a one × *n* vector. Each vector was multiplied by 255 and then transformed into a 24 × 24 matrix. This 24 × 24 matrix converted the image into a set of pixel values, whereby the localization image used in this study (Figure 6) was a single-channel grayscale map.

### 4.2. Room-Level Small-Scale Dataset Processing

Due to the limitations of the localization area and the similarity of RSSI data values in small-scale scenarios, insufficient data volume can hardly express the localization features of specific locations, even though data collection considers each localization point as much as possible. Therefore, this paper used a Wi-Fi fingerprint heatmap, which used different colors to represent the different RSSI values received by each localization point from the same transmitter. Such a heatmap could show noticeable differences and similarities between features and samples, expressing more representative elements of the localization points with less data.

#### Using Wireless InSite Simulation Dataset

Wireless InSite 3.3.0 is a software tool that can simulate the propagation of electromagnetic waves in complex indoor and outdoor environments. It can help with indoor positioning by providing accurate and realistic models of wireless channel characteristics, such as path loss, delay spread, angle of arrival, and received signal strength. Wireless InSite can also help design and optimize wireless systems and networks, such as Wi-Fi, 5G, IoT, and radar, by evaluating the performances and coverages of different antenna configurations, transmitter locations, and frequency bands. Because of the advantages of Wireless InSite, we used it to simulate the RSSI values of Wi-Fi signals propagated in real scenarios. The following is the specific method we used to construct the dataset.

The Wi-Fi fingerprint heatmap we construct was a fingerprint map of a single AP that identified the signal strengths of different receiving points at the localization point by color. The fingerprint heatmap of the localization point was formed by receiving RSSI values from 12 fixed transmitters in the localization area. The coordinates of the fixed transmitters in the same area were the same. In the image, only the signals of the fixed transmitters were considered, and other coordinate points were set as unselected WAPs. Figure 7 shows the fingerprint heatmaps of two different APs in the localization area. The *x*-axis and *y*-axis represent the coordinates of the transmitting points in the area, and the colors represent the magnitudes of the signal strengths of different APs at the transmitting points.

### 4.3. Improved Convolutional Neural Network Model

We propose a deep-learning model based on convolutional neural networks (CNNs) to predict user location from multinoise wireless signal strength fingerprint maps. Compared with traditional CNNs, our model performed better on the dataset constructed in this paper by improving the network structure and parameters. Considering the significant differences between the images with and without features in the fingerprint maps, we considered adding a ReLU function after the convolutional layer and dense layer to increase nonlinearity, filter out the values of the featureless areas, and improve the sparsity of the model. At the same time, we added separate convolutional structures before and after the dense layer to increase the extraction of compelling features multiple times. We called our model DS-CNN, which stands for deep signal-strength convolutional neural network. The DS-CNN consisted of four convolutional layers, four pooling layers, two fully connected layers, two dense layers, and a SoftMax layer, which output the indoor location probability. Figure 8 and Table 1 and Table 2 show a structural model diagram of the indoor location network, the output shape of each layer, and parameter settings for the proposed offline training dataset, respectively.

## 5. Simulation Analysis

### 5.1. Need for Simulation Environment

In indoor positioning algorithms and technologies, the accuracy and reliability of predicted positioning results are the most critical concerns. An indoor environment contains more uncertainties than an outdoor environment, with more obstructions that can cause unpredictable attenuation of signal strength during the propagation of wireless signals [37].

#### 5.1.1. Non-Line-of-Sight Propagation

Communication between devices generally involves a transmitter and a receiver, with the signal ideally directed along a straight line between the two points. However, there are many obstacles between a transmitter and receiver in an indoor environment, such as doors, windows, walls, and furniture. These obstacles can hinder signal refraction or reflection in the propagation process, resulting in a receiver not receiving accurate data. The blocking of signal attenuation by such indoor obstacles is called non-line-of-sight propagation.

#### 5.1.2. Multipath Propagation

Because multiple wireless signals are simultaneously present in a room during data transmission, indoor obstacles cause the numerous signs to reflect, scatter, and bypass; thus, the final signal received by a receiver is likely composed of the sum of multiple wireless signal strength vectors. This effect leads to a signal distortion effect called multipath propagation.

#### 5.1.3. Shadow Effect

Indoor environments typically comprise multiple rooms or studios for confidentiality in everyday activities. In these areas, signal attenuation can occur at the corners of walls, creating blind spots. As a result, the signal received by a receiver varies as it moves. This variation in signal strength is known as the shadowing effect.

In order to verify the impact of the aforementioned influencing factors on front-end data acquisition, we conducted multiple measurements of four access points (APs) at the field site at different times of the day. The measurement results are shown in Figure 9 below. The measurement results were influenced by the multipath effect caused by the activities of personnel and nonvisual propagation at the field site. The RSSI value of an AP fluctuated significantly at different times, causing the RSSI data values collected at the front end to vary significantly. Thus, the constructed offline dataset did not have localization characteristics.

### 5.2. Simulation Environment Setup

The small-scale natural environments were rooms 407, 512, and 308 of the School of Electrical Information of Wuhan Institute of Technology and the corridor area.

The simulated walking step was assumed to be 0.5 m. A total of 836 points and 8312 data records were manufactured and collected. Figure 10 and Figure 11 show illustrations of the 3D scene simulated with Wireless InSite and the actual scene in the experiment. In the simulation scene, TX represents the transmitter’s position, and the receiving points are distributed throughout the area at an interval of 0.5 m. Detailed simulation parameter settings are shown in Table 3.

To make the simulated scene as close as possible to the location of the actual room, the sizes and makeups of the items in the room were manufactured to match the path loss incurred in the virtual space. However, in practical scenarios, the materials used for buildings are too complex and the materials available in simulation software are too simple, which makes it impossible to simulate accurately. Moreover, the human activities in the laboratory and the temperature changes throughout the day caused the difference between the actual measurement and the simulation to be within the 3–4 dBm range. As shown in Figure 12, we deployed the same transmitter and receiver in the natural and simulated scenarios. We collected the RSSI values from the same transmitter at a distance of 15 m. The results show that the trend of the actual measured RSSI values was clearly below the simulated values.

### 5.3. Accuracy Comparison of Different Algorithms for the UJIIndoorLoc Database

This section compares the accuracy of the proposed method using the UJIIndoorLoc database with the already existing depth model indoor localization framework of CNNLoc and others (shown in Figure 13) to verify that the proposed network outperformed other networks in terms of indoor accuracy. The CNNLoc network was proposed by Song et al. [38] for multibuilding and multifloor indoor localization. Our proposed DS-LocCNN framework was compared with CNNLoc, a traditional CNN, KNN, and an SVM [39,40] concerning the average floor accuracy using the UJIIndoorLoc dataset. As shown in Table 4, the best floor accuracy of the well-performing CNNLoc was 96.03%, and the highest accuracy of the proposed method for floor localization was 96.67%, higher than the well-performing CNNLoc and other localization methods.

In order to verify the iterative accuracy performance of the proposed method on the UJIIndoorLoc dataset, validation experiments were conducted using traditional CNN and MobileNet models. Validation experiments were performed using a conventional CNN and MobileNet. Figure 14 shows plots of the accuracy and loss of the proposed method for 30 iterations of training and validation sets from the UJIIndoorLoc dataset. Figure 15 shows the accuracies of the traditional CNN and MobileNet for 30 iterations of training and validation sets from the UJIIndoorLoc dataset. Table 5 records the best performances of the three methods over 30 iterations.

The UJIIndoorLoc dataset was used as an experimental object in large-scale scenarios. Although it had a rich representation of each raw datapoint and clear differences between different WAPs, the signal strength from the same transmitter in the same area was similar to the fingerprint map, which did not show an advantage in simple feature extraction. Therefore, the performances of MobileNet and CNN were not satisfactory.

### 5.4. Accuracy Comparison of Different Algorithms for the Simulated Dataset

Validation experiments were conducted using traditional CNN and MobileNet models to verify the proposed method’s performance on the simulated dataset with iterative accuracy. Figure 16 shows the accuracy and loss of the proposed method for the simulated dataset with 30 iterations of training and validation sets. Figure 17 shows the accuracies of the traditional CNN and MobileNet for the simulated dataset with 30 iterations of training and validation sets. The best performances of the three methods for the simulated Wireless Insite dataset are shown in Table 6. The proposed method in this paper achieved an accuracy of 99.06% on the validation set, which was better than the performances of either the traditional CNN or MobileNet.

Undeniably, the location of adjacent APs in indoor positioning significantly impacts positioning accuracy and limits a network model’s performance. Compared with large-scale scenarios, small-scale scenarios had the advantage of adding heat maps to process front-end datasets. As the results show, this method of enhancing the specific area features of a fingerprint map was effective.

### 5.5. Accuracy Comparison of Different Algorithms for the Measured Dataset

Validation experiments were conducted using a traditional CNN and MobileNet to verify the proposed method’s performance on the measured dataset with iterative accuracy. Figure 18 shows the accuracy and loss of the proposed method for the measured dataset with 30 iterations of training and validation sets. Figure 19 shows the accuracies of the traditional CNN and MobileNet for the measured dataset with 30 iterations of training and validation sets. The best performances of the three methods for the measured dataset are shown in Table 7. The proposed method in this paper achieved an accuracy of 99.10% for the validation set, which was better than the performances of either the traditional CNN or MobileNet.

## 6. Conclusions

In this study, we proposed a neural-network-based localization method for Wi-Fi fingerprint indoor localization. Our approach considered the needs of localization areas of different scales. We processed the acquired raw data to construct a grayscale fingerprint map for large-scale scenarios and a thermal fingerprint map for small-scale settings, which could better fit the training requirements of the corresponding scenario data. We conducted experiments on both simulated and real datasets, and the results showed that our proposed method could achieve over 99% validation accuracy for both. Our approach could reduce the workload of front-end data collection to some extent and also provided data support for algorithm validation in localization research.

Similarly, this study still needs to be continuously optimized in future work.

1. The existing method only considered RSSI values in ideal environments when simulating and measuring the positioning data. Since RSSI values are very sensitive to devices and environments, future work should aim to reduce the impacts of uncertain factors on measurement, improve the model’s compatibility with RSSI value fluctuations, and further optimize its performance in positioning.

2. The existing method was based on convolutional neural networks for positioning result classification and discussed the location results in small-scale and large-scale scenarios. Future work should be more fine-grained, consider point-level location output, and optimize the model for more accurate location positioning.

## Figures and Tables

**Figure 1 sensors-23-06992-f001:**
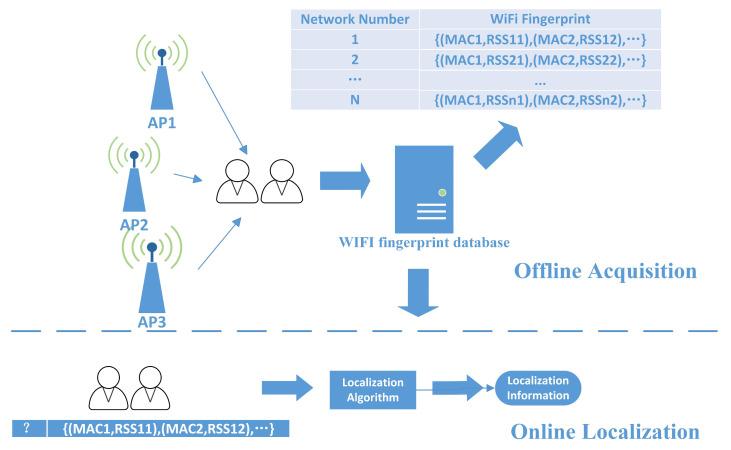
Traditional indoor positioning process based on fingerprinting technology.

**Figure 2 sensors-23-06992-f002:**
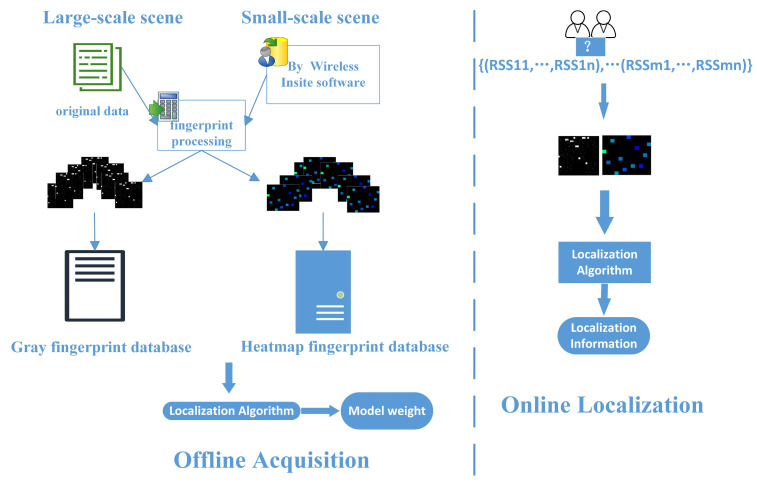
Proposed indoor fingerprint-based localization process.

**Figure 3 sensors-23-06992-f003:**
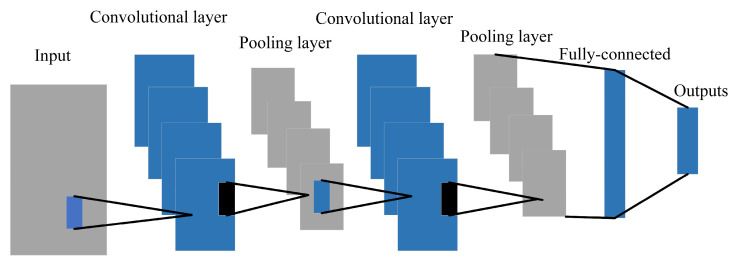
Structure of a convolutional neural network.

**Figure 4 sensors-23-06992-f004:**
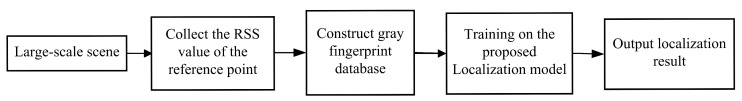
Indoor localization process for large-scale scenarios.

**Figure 5 sensors-23-06992-f005:**
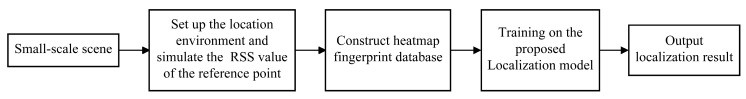
Indoor localization process for small-scale scenarios.

**Figure 6 sensors-23-06992-f006:**
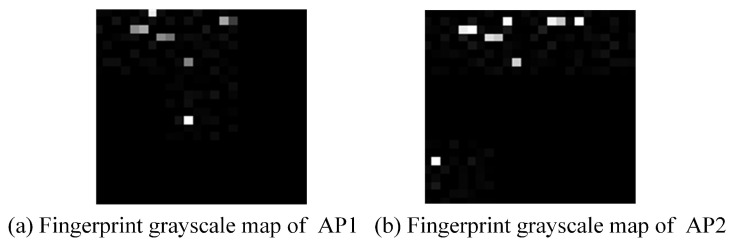
Fingerprint grayscale maps of two different access points in the localization area.

**Figure 7 sensors-23-06992-f007:**
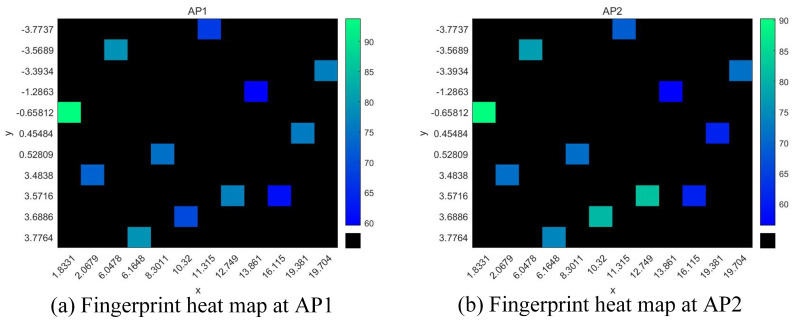
Fingerprint heatmaps of two different access points in the localization area.

**Figure 8 sensors-23-06992-f008:**
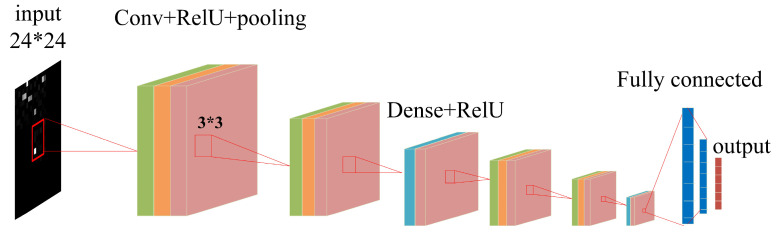
Structure of proposed model of an indoor positioning network for training offline dataset.

**Figure 9 sensors-23-06992-f009:**
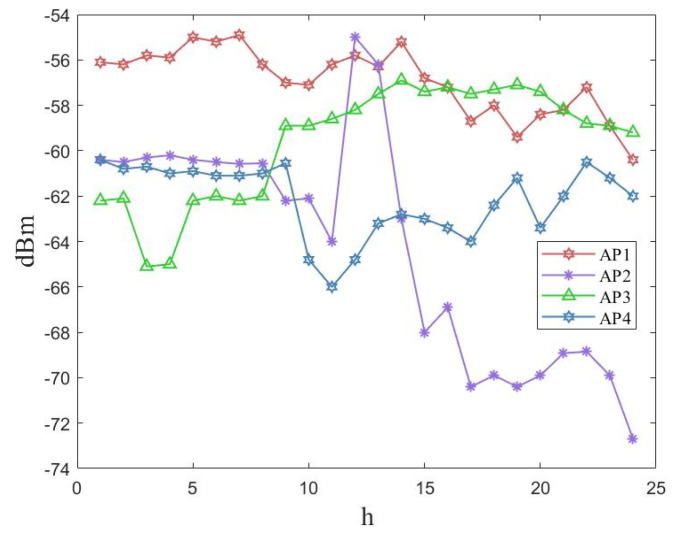
RSSI changes at different APs throughout the day.

**Figure 10 sensors-23-06992-f010:**
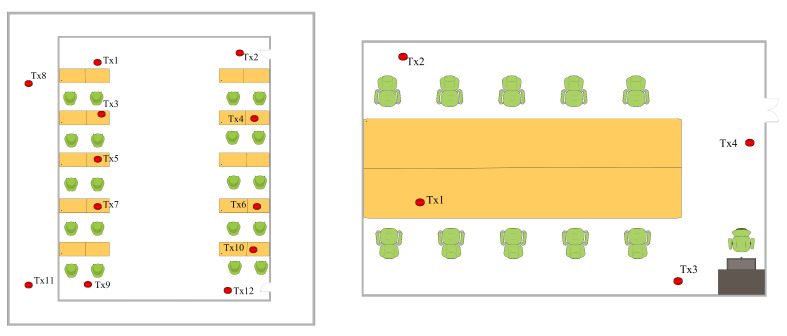
Simulated maps of rooms 407 (**left**) and 512 (**right**).

**Figure 11 sensors-23-06992-f011:**
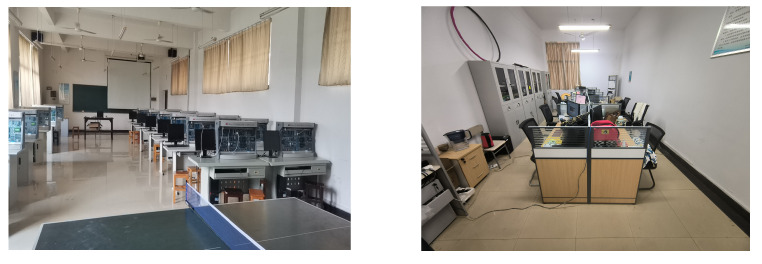
Real images of rooms 407 (**left**) and 512 (**right**).

**Figure 12 sensors-23-06992-f012:**
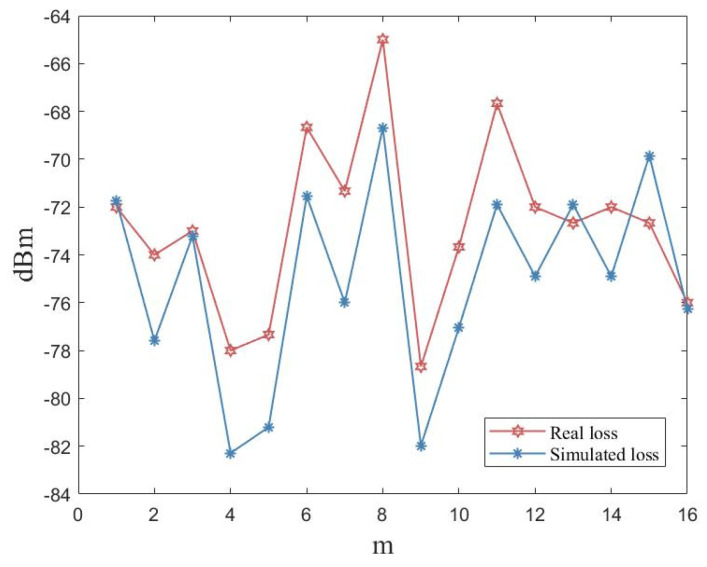
Path loss in real and simulated scenarios.

**Figure 13 sensors-23-06992-f013:**
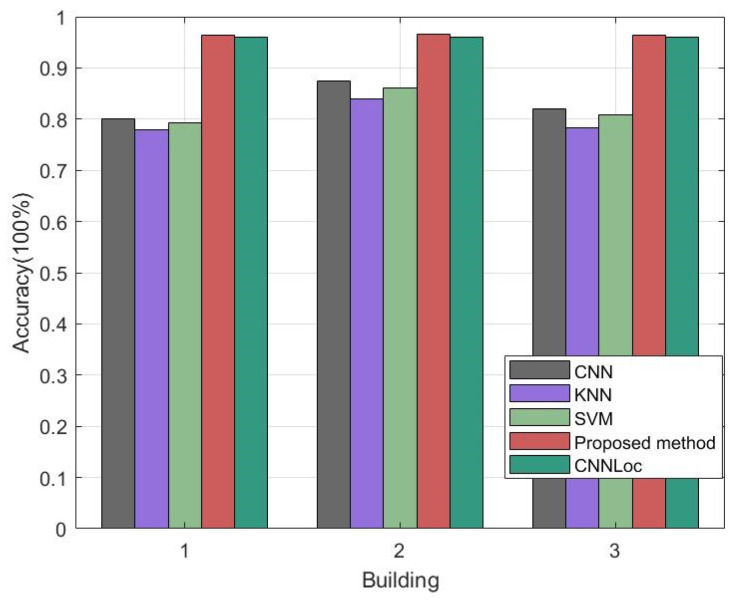
Average floor accuracy of each localization method with the UJIIndoorLoc datasets.

**Figure 14 sensors-23-06992-f014:**
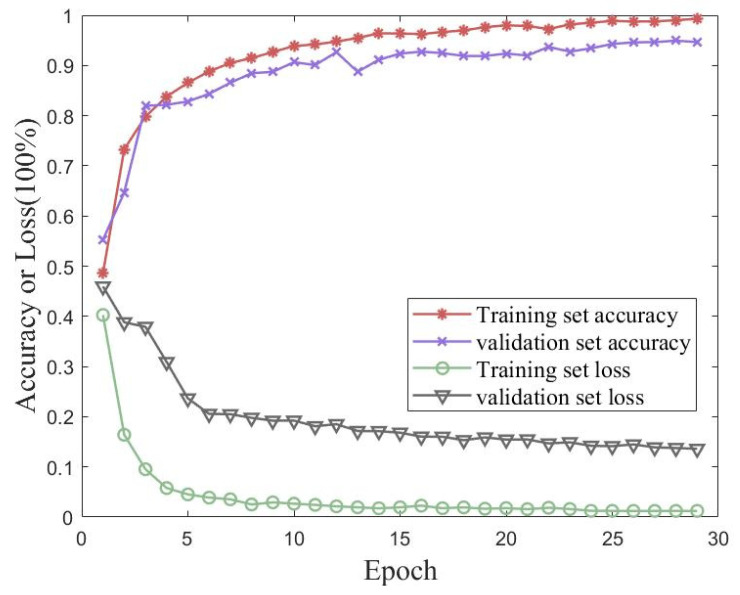
Accuracy and loss of the proposed method for 30 iterations of training and validation sets from the UJIIndoorLoc dataset.

**Figure 15 sensors-23-06992-f015:**
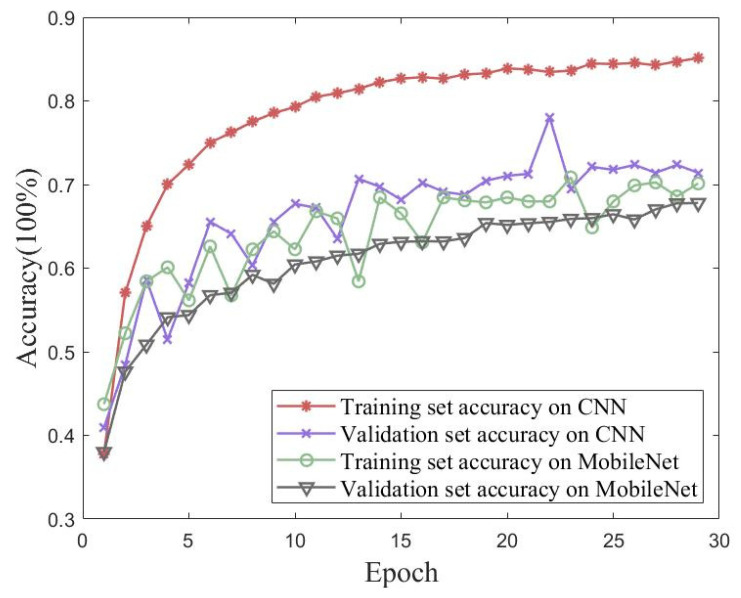
Accuracy of traditional CNN and MobileNet models for 30 iterations of training and validation sets from the UJIIndoorLoc dataset.

**Figure 16 sensors-23-06992-f016:**
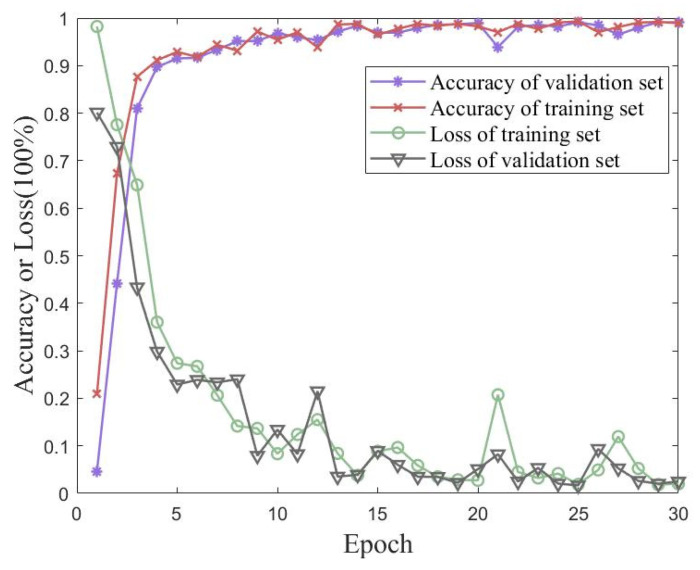
Accuracy and loss of the proposed method for 30 iterations of training and validation sets with the simulated dataset.

**Figure 17 sensors-23-06992-f017:**
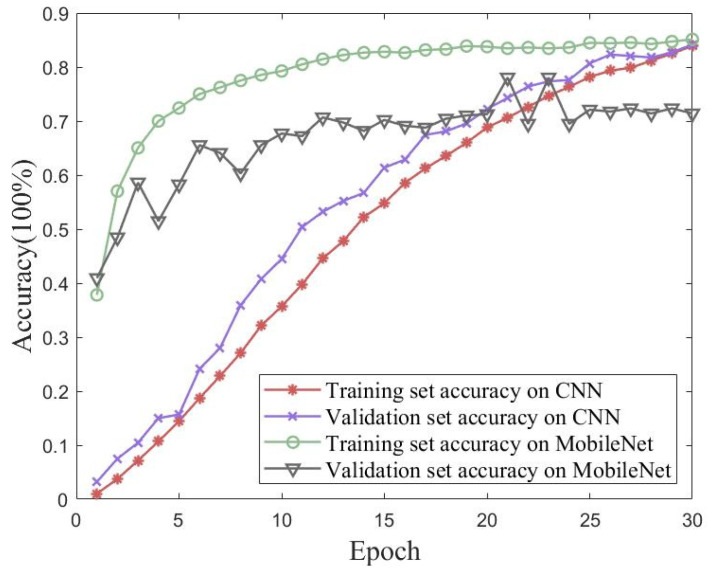
Accuracies of the traditional CNN and MobileNet for 30 iterations of training and validation sets with the simulated dataset.

**Figure 18 sensors-23-06992-f018:**
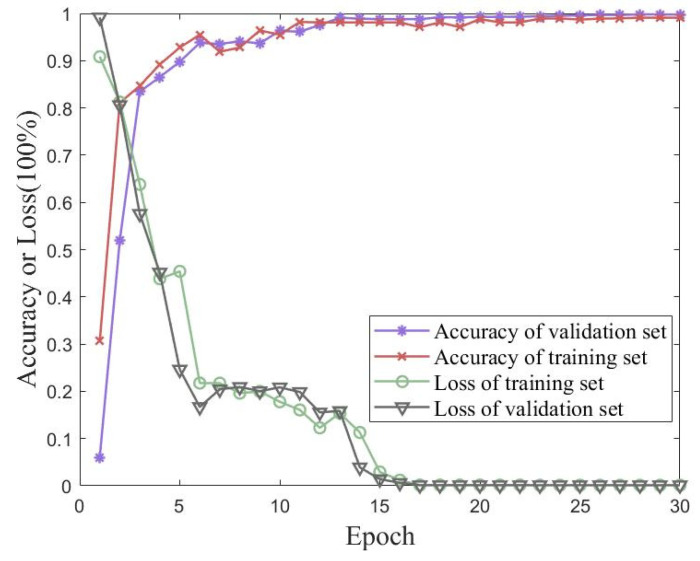
Accuracy and loss of the proposed method for 30 iterations of training and validation sets with the measured dataset.

**Figure 19 sensors-23-06992-f019:**
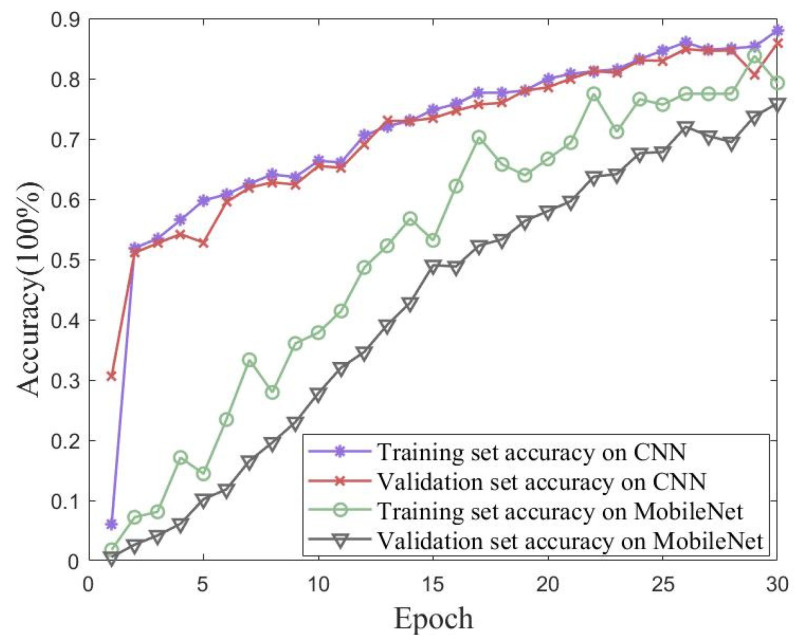
Accuracies of the traditional CNN and MobileNet for 30 iterations of training and validation sets with the measured dataset.

**Table 1 sensors-23-06992-t001:** The output shape of each layer in our method.

Layer	Parameter	Output Shape
Input	Training data	(224, 224, 3, m)
Conv2D 1	Conv2D	(222, 222, 32, m)
Conv2D 2	Conv2D	(109, 109, 64, m)
Conv2D 3	Conv2D	(52, 52, 32, m)
Conv2D 4	Conv2D	(24, 24, 64, m)
MaxPooling1	MaxPooling2D	(111, 111, 32, m)
MaxPooling2	MaxPooling2D	(54, 54, 64, m)
MaxPooling3	MaxPooling2D	(26, 26, 32, m)
MaxPooling4	MaxPooling2D	(12, 12, 64, m)
Flatten	K = 0	(9216, m)
dense1	K = 1,179,776	(128, m)
dense2	K = 516	(4, m)
Output	K = Nrp	(N, m)

**Table 2 sensors-23-06992-t002:** Parameter settings for our method.

Parameter	Value
Convolutional layers	4
Pooling	4
Pooling size	2 × 2
Stride	1
Dense layers	2
Fully connected layers	2
SoftMax layer	1
Kernel	3 × 3

**Table 3 sensors-23-06992-t003:** Parameter settings for Wireless Insite.

Layer	407 Value	512 Value
Length × width × height/m	17.5 × 8 × 3.5	7.5 × 4 × 3.5
Transmit signal frequency/GHz	5	
Transmit power/dBm	23	
Transmit antenna height/m	1	
Receive antenna height/m	1	
Type of transmit antenna	half-wave dipole	
Type of receive antenna	isotropic	
Polarization form of transmit antenna	vertical	
Polarization form of receiving antenna	vertical	
Sampling interval at the receiver/m	0.5	
Noise figure of receiver/dBm	3	
Transmit signal frequency/GHz	5	

**Table 4 sensors-23-06992-t004:** Average building accuracy of different algorithms with the UJIIndoorLoc datasets.

Method	First Building/%	Second Building/%	Third Building/%
Ours	96.32	96.67	96.41
CNNLoc	96.03	96.03	96.03
CNN	80.13	87.41	82.04
SVM	79.29	86.01	80.90
KNN	77.99	84.01	78.29

**Table 5 sensors-23-06992-t005:** Performances of different algorithms with the UJIIndoorLoc dataset.

Method	Training Accuracy/%	Validation Accuracy/%	Training Loss/%	Validation Loss/%
Ours	99.54	96.41	0.0123	0.1360
MobileNet	70.85	67.79	1.8511	1.7125
CNN	85.12	72.35	0.9302	0.8798

**Table 6 sensors-23-06992-t006:** Performances of different algorithms for the simulated Wireless Insite dataset.

Method	Training Accuracy/%	Validation Accuracy/%	Training Loss/%	Validation Loss/%
Ours	99.12	99.06	0.0185	0.0168
MobileNet	84.18	83.90	0.9372	0.8764
CNN	85.12	78.05	0.9263	0.8945

**Table 7 sensors-23-06992-t007:** Performances of different algorithms for the measured dataset.

Method	Training Accuracy/%	Validation Accuracy/%	Training Loss/%	Validation Loss/%
Ours	99.78	99.10	0.0011	0.0012
CNN	88.00	85.86	0.8635	0.8964
MobileNet	83.78	75.89	1.8610	1.7307

## Data Availability

No new data were created or analyzed in this study. Data sharing is not applicable to this article.

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
