# Peer review of "Neural-Network-Based Localization Method for Wi-Fi Fingerprint Indoor Localization"

_sensors, 2023, doi:10.3390/s23156992_

Round 1
Reviewer 1 Report
Comments to the Author
This paper proposes a fingerprinting technology based on CNNs to improve Wifi indoor localisation and proposes a new indoor localisation model using 3D ray-tracing techniques to simulate wireless RSSI in the field area. It is an interesting research topic and indoor localisation has many potential application areas. However, there are several points that need to be addressed to improve the quality of the manuscript.
Suggestions to improve the quality of the paper are provided below:
1. The authors should include a few examples in the Introduction section of the real-world applications of indoor localization systems. Indoor positioning is leveraged intensely, especially in the building domain where several popular real-world applications are worth mentioning to target a large audience. These building applications include emergency management, smart energy management and HVAC controls and occupancy detection. Please review the following established works as a good starting point to highlight the important applications where indoor positioning leveraged.
Indoor localisation for building emergency management
10.1109/IUCC-CSS.2016.013
Indoor localisation for smart energy management
https://doi.org/10.1016/j.buildenv.2022.109472
Indoor localisation for smart HVAC controls
https://doi.org/10.1145/2517351.2517370
Indoor localisation for occupancy prediction
https://doi.org/10.1016/j.buildenv.2022.109689
2. Similarly, the authors should also include examples of few real-world applications of GPS-based location services, such as vehicle activity recognition, navigation, and activity recommendation. Please review the following established works as a good starting point to highlight these applications.
Vehicle activity recognition
10.1109/TITS.2020.2970229
Vehicle navigation
10.1109/5.736347
Location and activity recommendation
https://doi.org/10.1145/1772690.1772795
3. Given that Wifi and Bluetooth Low Energy are two of the most common wireless technologies used for indoor localisation, please refer to the following paper [1] as a good starting point and provide a short discussion comparing between both of these technologies. Then, explain why Wifi technology was chosen for this study.
[1] https://doi.org/10.1016/j.buildenv.2020.106681
4. Please provide a reason why the dBm values for simulated loss is consistently lower than the dBm values for real loss.
5. The descriptions for the UJIndoorLoc dataset and Wireless Insite need to be more clearer explained with their own subsection headers.
6. I am unsure about the purpose of the analysis conducted in Page 12 and 13 on verifying the iterative accuracy performance when using a traditional CNN and MobileNet. Why is the proposed approach and CNNLoc not included in this analysis?
7. The authors should make some attempt to explain the reasoning behind the proposed approach’s superior performance over other approaches, instead of simply showing the experimental results.
8. Please include a discussion on the limitations of the existing approach and elaborate on how they can be improved in future works. For instance, the performance difference between CNNLoc and the proposed solution is less than 1%. How will this be further improved in the future? How is the proposed solution better than CNNLoc in other aspects?
9. Minor comments
· Table 1 can be improved by also including information about the window size and stride size used in the convolutional layers.
· Figure 9 needs to be improved by standardising the use of colours and including a legend to describe the icons used in the figure.
· Figure 10 seems to be showing the same image for both rooms 407 and 512.
· Table 3 can be improved by sorting the results in descending order.

There are only minor issues with the manuscript's quality of English, which have been listed in the comments section.
Author Response
Dear reviewer,
We are very grateful for your comments and professional suggestions. These views help to improve the academic rigor of our paper. According to your suggestions and requirements, we have revised and corrected the modified manuscript, and we hope that our work can be improved again. In addition, we would like to explain in detail as follows:
- Introduction: The authors should include a few examples in the Introduction section of the real-world applications of indoor localization systems. Indoor positioning is leveraged intensely, especially in the building domain where several popular real-world applications are worth mentioning to target a large audience. These building applications include emergency management, smart energy management and HVAC controls and occupancy detection. Please review the following established works as a good starting point to highlight the important applications where indoor positioning leveraged.
Indoor localisation for building emergency management
10.1109/IUCC-CSS.2016.013
Indoor localisation for smart energy management
https://doi.org/10.1016/j.buildenv.2022.109472
Indoor localisation for smart HVAC controls
https://doi.org/10.1145/2517351.2517370
Indoor localisation for occupancy prediction
https://doi.org/10.1016/j.buildenv.2022.10968
- Introduction: Similarly, the authors should also include examples of few real-world applications of GPS-based location services, such as vehicle activity recognition, navigation, and activity recommendation. Please review the following established works as a good starting point to highlight these applications.
The author's answer: Following your suggestion, we have added the established works that you suggested in the introduction section, and improved the citation, to highlight the important applications and good starting points of indoor localization.
- Introduction:Given that Wifi and Bluetooth Low Energy are two of the most common wireless technologies used for indoor localisation, please refer to the following paper [1] as a good starting point and provide a short discussion comparing between both of these technologies. Then, explain why Wifi technology was chosen for this study.
[1] https://doi.org/10.1016/j.buildenv.2020.106681
The author's answer: Regarding these two technologies, WIFI and Bluetooth low energy, we have briefly discussed them in line 72 of the paper. In RSSI-based indoor localization, Wi-Fi and Bluetooth are the two most common wireless technologies used indoors. In Bluetooth low energy-based indoor localization, in short, these two indoor localization schemes are very similar in terms of the localization process, except for the different wireless technologies used. In this paper, we choose WIFI technology for research considering that in practical applications, Wi-Fi devices are ubiquitous in real scenarios, including routers, smartphones, etc. in the laboratory. In order to meet the number of sending and receiving devices required in the data collection stage, and also to minimize the cost of funds, we choose to use Wi-Fi technology for this research. If we need to conduct more in-depth research on specific scenarios for Bluetooth in the future, our team will also consider using Bluetooth low energy to conduct more in-depth research.
- Introduction: Please provide a reason why the dBm values for simulated loss is consistently lower than the dBm values for real loss.
The author's answer: In this paper, we used Wireless InSite, a simulation software that can be applied to analyze the radio wave propagation and wireless communication system performance characteristics of individual sites, when simulating the scenarios. However, in the real scenarios, the materials used in the buildings are too complex, and the materials that can be selected in the simulation software are too simple, which cannot accurately simulate the real situations. Moreover, due to the personnel activities in the laboratory and the temperature changes throughout the day, the actual measured dBm is always lower than the ideal conditions. In our future research, we will also specifically address this issue and conduct more in-depth exploration, using more accurate measurement data to reduce the impact of dBm values on localization.
- Introduction: The descriptions for the UJIndoorLoc dataset and Wireless Insite need to be more clearer explained with their own subsection headers.
The author's answer: We added subheadings in sections 4.1.1 and 4.2.1 to describe the UJIndoorLoc dataset and Wireless Insite, respectively.
- Introduction: I am unsure about the purpose of the analysis conducted in Page 12 and 13 on verifying the iterative accuracy performance when using a traditional CNN and MobileNet. Why is the proposed approach and CNNLoc not included in this analysis?
The author's answer: We apologize for our mistake in editing the paper, which caused the wrong information of the figures. We have corrected it now. Figure 14 shows the accuracy and loss curves of our proposed method on the UJIIndoorLoc dataset. Since CNNLoc was proposed by Song et al, they only showed the average floor accuracy on the UJIIndoorLoc dataset in their paper. Due to the unavailability of their code, we could not accurately know their best performance in 30 epochs.
- Introduction: The authors should make some attempt to explain the reasoning behind the proposed approach’s superior performance over other approaches, instead of simply showing the experimental results.
The author's answer: We have improved the description of the experimental results in the paper, and explained the reasons why our proposed method outperforms other methods.
- Introduction: Please include a discussion on the limitations of the existing approach and elaborate on how they can be improved in future works. For instance, the performance difference between CNNLoc and the proposed solution is less than 1%. How will this be further improved in the future? How is the proposed solution better than CNNLoc in other aspects?
The author's answer: 1. Existing methods only consider the RSS values under ideal conditions when simulating and measuring localization data. Since RSS values are very sensitive to devices and environments, future work should aim to reduce the impact of uncertainties on measurements and improve the model's compatibility with RSS value fluctuations. 2. Existing methods are based on convolutional neural networks to classify the localization results, and discuss the location results in small-scale and large-scale scenarios. Future work should be more refined, considering location outputs above the point level, and also further optimize the deep learning model, to make more accurate location localization serve the high-precision requirements. In response to the above answers, we have discussed in the paper how future work can improve the limitations of existing methods.
- Introduction: Minor comments
1.Table 1 can be improved by also including information about the window size and stride size used in the convolutional layers.
- Figure 9 needs to be improved by standardising the use of colours and including a legend to describe the icons used in the figure.
3.Figure 10 seems to be showing the same image for both rooms 407 and 512.
4.Table 3 can be improved by sorting the results in descending order.
The author's answer: We have revised the issues of Table 1, Figure 9, Figure 10 and Table 3.
Reviewer 2 Report
Although the article proposed a convolutional neural network to improve Wi-Fi indoor localization and a new indoor localization model, it has many unsatisfied points. The followings are the suggestions and comments.
1. It looks that the article just brought out an indoor localization flow of indoor localization.
In large-scale, some procedures, such as construct of fingerprint grayscale datasets and convolutional neural network, are already common approaches. Refer to the following literatures,(1) Sun, Danshi, et al. "Improving fingerprint indoor localization using convolutional neural networks." IEEE Access 8 (2020): 193396-193411.(2) Liu, Zhenyu, et al. "Hybrid wireless fingerprint indoor localization method based on a convolutional neural network." Sensors 19.20 (2019): 4597.
In small-scale, there is no doubt that the RSSI model is widely used in many localization methods and some improved/ calibrated model have been proposed. Refer to the following literatures, (1) Sinha, Rashmi Sharan, and Seung-Hoon Hwang. "Improved RSSI-based data augmentation technique for fingerprint indoor localisation." Electronics 9.5 (2020): 851.(2) Li, Guoquan, et al. "Indoor positioning algorithm based on the improved RSSI distance model." Sensors 18.9 (2018): 2820.
On the whole, it is recommended that the specific contributions of the article should be highlighted and listed, which are not clear in the introduction.
2. The proposed flow separates large-scale and small-scale procedures . But what is the standard of these two scales? It is suggested that the principle of scenario differentiation should be clearly stated.
3. In section 4.3, it is not clear of the improved points of the proposed Convolutional Neural Network Model. It seems that the CNN includes multiple traditional layers without any further modification. And some CNNs were proposed , such as the references of question 1.
4. In the experiments, it also has many unsatisfactory points. (1) The comparison of floor and building accuracy using UJIIndoorLoc dataset is unsatisfied. The floor and building location are not tough tasks in this dataset. Some methods can achieve almost 100% accuracy, such as in the literatures, Kim, Kyeong Soo, Sanghyuk Lee, and Kaizhu Huang. "A scalable deep neural network architecture for multi-building and multi-floor indoor localization based on Wi-Fi fingerprinting." Big Data Analytics 3 (2018): 1-17. In this article, the building hit rates are all above 99%. It is recommended that some other location indexes, such as location distance, should be considered.(2) It is no doubt that KNN and SVM are classical algorithms in indoor localization. But many improvements of these methods have been proposed. And the advanced Convolutional Neural Networks for location have been brought out in recent years. So it is suggested to compare the proposed method with the newest algorithm.
5.The purpose of the article is to solve the problem of a significant amount of time and labor in the collection of data. It is recommended that the analysis of labeled data and localization accuracy should be provided. Furthermore, it is better to compare proposed methods with other algorithms in different size of labeled data which proof that the proposed methods with small data size or light labor can achieve statisfied accuracy.
6. Please check the mistakes of article, there are obvious clerical errors in several places. Such as “110m2”, strange symbols before “104 dBm” and “110 dBm”
The English writing can be acceptable.
Author Response
Dear reviewer,
We are very grateful for your comments and professional suggestions. These views help to improve the academic rigor of our paper. According to your suggestions and requirements, we have revised and corrected the modified manuscript, and we hope that our work can be improved again. In addition, we would like to explain in detail as follows:
- Introduction: It looks that the article just brought out an indoor localization flow of indoor localization.
In large-scale, some procedures, such as construct of fingerprint grayscale datasets and convolutional neural network, are already common approaches. Refer to the following literatures,(1) Sun, Danshi, et al. "Improving fingerprint indoor localization using convolutional neural networks." IEEE Access 8 (2020): 193396-193411.(2) Liu, Zhenyu, et al. "Hybrid wireless fingerprint indoor localization method based on a convolutional neural network." Sensors 19.20 (2019): 4597.
In small-scale, there is no doubt that the RSSI model is widely used in many localization methods and some improved/ calibrated model have been proposed. Refer to the following literatures, (1) Sinha, Rashmi Sharan, and Seung-Hoon Hwang. "Improved RSSI-based data augmentation technique for fingerprint indoor localisation." Electronics 9.5 (2020): 851.(2) Li, Guoquan, et al. "Indoor positioning algorithm based on the improved RSSI distance model." Sensors 18.9 (2018): 2820.
On the whole, it is recommended that the specific contributions of the article should be highlighted and listed, which are not clear in the introduction.
The author's answer: We have highlighted and listed the specific contributions of the paper in lines 129 to 143 of the introduction. In summary, (1) Sun, Danshi, et al. "Improving fingerprint indoor localization using convolutional neural networks." IEEE Access 8 (2020): 193396-193411. (1) proposed to convert the collected bluetooth RSS to the fingerprint images required for computation, and established a CNN for classification training, but the data set used was from actual measurements, which consumed a lot of effort in the front-end collection. (2) Liu, Zhenyu, et al. "Hybrid wireless fingerprint indoor localization method based on a convolutional neural network." Sensors 19.20 (2019): 4597. (3) Sinha, Rashmi Sharan, and Seung-Hoon Hwang. "Improved RSSI-based data augmentation technique for fingerprint indoor localisation." Electronics 9.5 (2020): 851. (2) proposed to construct ratio fingerprints by calculating the ratio of different RSS values of important contribution access points. (3) simulated the changing RSS values based on the reference RSS values, achieving data augmentation. If the scenario requirements can be divided, the computation of important contribution access points and the complexity of data augmentation can also be reduced or lowered accordingly. (4) Li, Guoquan, et al. "Indoor positioning algorithm based on the improved RSSI distance model." Sensors 18.9 (2018): 2820. (4) proposed to upload the RSSI fluctuations of the detected bluetooth nodes to the cloud, and correct the RSSI values in real time, but the cost of the cloud server also increases with the increase of data.
- Introduction: The proposed flow separates large-scale and small-scale procedures . But what is the standard of these two scales? It is suggested that the principle of scenario differentiation should be clearly stated.
The author's answer: The criterion for distinguishing between large-scale and small-scale scenarios is whether to perform room point-level or building floor-level localization, and to choose the corresponding process according to the localization accuracy requirements of these two scenarios. We have improved the description of this issue in lines 256 to 269 of section 4 of the paper.
- Introduction: In section 4.3, it is not clear of the improved points of the proposed Convolutional Neural Network Model. It seems that the CNN includes multiple traditional layers without any further modification. And some CNNs were proposed , such as the references of question 1.
The author's answer: We have added the improvement of our method on convolutional neural networks in Section 4.3, and also added the description of parameter settings.
- Introduction: In the experiments, it also has many unsatisfactory points. (1) The comparison of floor and building accuracy using UJIIndoorLoc dataset is unsatisfied. The floor and building location are not tough tasks in this dataset. Some methods can achieve almost 100% accuracy, such as in the literatures, Kim, Kyeong Soo, Sanghyuk Lee, and Kaizhu Huang. "A scalable deep neural network architecture for multi-building and multi-floor indoor localization based on Wi-Fi fingerprinting." Big Data Analytics 3 (2018): 1-17. In this article, the building hit rates are all above 99%. It is recommended that some other location indexes, such as location distance, should be considered.(2) It is no doubt that KNN and SVM are classical algorithms in indoor localization. But many improvements of these methods have been proposed. And the advanced Convolutional Neural Networks for location have been brought out in recent years. So it is suggested to compare the proposed method with the newest algorithm.
The author's answer: We added an experiment at the end of the experimental validation section to supplement the performance of our method on the real-world dataset, and compared it with other algorithms. Since the algorithm comparison requires the same dataset and the data support from other authors, we only compared our method with a few methods mentioned in the paper using the same variable. In our future research, we will also strive to solve this problem, so that our method can achieve better performance in the existing field.
- Introduction: The purpose of the article is to solve the problem of a significant amount of time and labor in the collection of data. It is recommended that the analysis of labeled data and localization accuracy should be provided. Furthermore, it is better to compare proposed methods with other algorithms in different size of labeled data which proof that the proposed methods with small data size or light labor can achieve statisfied accuracy.
The author's answer: We added an experiment at the end of the experimental validation section to supplement the performance of our method on the real-world dataset, and compared it with other algorithms. Since the algorithm comparison requires the same dataset and the data support from other authors, we only compared our method with a few methods mentioned in the paper using the same variable. In our future research, we will also strive to solve this problem, so that our method can achieve better performance in the existing field.
- Introduction: Please check the mistakes of article, there are obvious clerical errors in several places. Such as “110m2”, strange symbols before “104 dBm” and “110 dBm”
The author's answer: We have fixed the strange symbols before "110m2", "104dbm" and "110dbm".
Reviewer 3 Report
1. The usage of translation software is good, but the indiscriminate usage of translation software is bad. The paper is littered with traces of translation software, and the author didn’t check that the translated sentences are smooth and correct.
2. There are many poor explanations in the paper:
① In Section 3.2, the convolutional neural network model consists of an input layer, stacked convolutional layers, downsampling layers, a fully connected layer, and an output layer, but the later part of the paper just describes the convolutional layer, pooling layer, and fully connected layer, without explaining the relationship between the downsampling layer and the pooling layer.
② In Section 4.1, the paper does not present the subscript km of RSSInkm, and does not present the subscript new and superscript k of RSSIn,km,new in Eq. (3).
③ In lines 247-248, there is no explanation of how a 1 × n vector is converted to a 24 × 24 matrix when multiplied by 255.
④ In line 268, the paper does not describe the origin of DS-CNN.
3. The usage of appropriate pictures in the paper can help readers better understand the content of the paper, but the pictures must be clear enough. Figures 1 and Figure 2 in the paper have very low clarity and it is difficult to read the text in them.
4. There are many errors of detail in the article:
① Equation (1) is inconsistent with the contents.
② The structure of a CNN described in Section 3.2 is inconsistent with Figure 3
③ In line 240, n is not the index of fingerprint collected in the mth acquisition point, n is independent of m.
④ Figure 10 shows real images of rooms 407 and 512, but the two images are identical.
⑤ In Table 2, the last two rows of parameters are clearly duplicated in the first two rows.
⑥ In Figure 13, there is only a loss curve, not an accuracy curve, which is inconsistent with the content.
⑦ In line 372, “Table Ⅳ” should be replaced by “table 4”.
5. There are also many problems with the experimental proof section of the article:
① The results presented in Table 4 are inconsistent with the content.
② The results presented in Table 4 are quite different from those presented in Figure 13 and Figure 14.
③ Iteration curves for the accuracy and loss values of the proposed method on the UJIIndoorLoc dataset are not shown in the article.
④ In lines 379-384, the author only gives the experimental data, does not present the data in a table, analyze the data in any way, or draw any relevant conclusions.
The traces of the usage of translation software in the article are extremely serious, and the author has not meticulously checked the article after translation, resulting in inconsistencies and grammatical errors in many parts of the article:
1. In line 178, “expressed” should be replaced by “measured”.
2. In line 183, the definite article “the” should be added before "distance" and described "the distance" in detail.
3. The terms referred to in lines 285 and 292 should both be “Non-line-of-sight Propagation", but the authors use different terms.
4. method column in Table 3, the method proposed in this paper should use “Ours”, not “Our”.
5. The authors use passive voice extensively, even in the contributions section, which is bad.
6. Authors like to use “this study” as a substitute for themselves, for example in line 159, “this study improves the ...” but the real subject should not be “this study” but “we”. the sentence should be replaced by “we improve the ... in this study.”
7. There is improper use of the singular and plural in the article.
Author Response
Dear reviewer,
We are very grateful for your comments and professional suggestions. These views help to improve the academic rigor of our paper. According to your suggestions and requirements, we have revised and corrected the modified manuscript, and we hope that our work can be improved again. In addition, we would like to explain in detail as follows:
- Introduction: The usage of translation software is good, but the indiscriminate usage of translation software is bad. The paper is littered with traces of translation software, and the author didn’t check that the translated sentences are smooth and correct.
The author's answer: We have rechecked the whole paper and corrected the sentences according to the requirements.
- Introduction: There are many poor explanations in the paper:
① In Section 3.2, the convolutional neural network model consists of an input layer, stacked convolutional layers, downsampling layers, a fully connected layer, and an output layer, but the later part of the paper just describes the convolutional layer, pooling layer, and fully connected layer, without explaining the relationship between the downsampling layer and the pooling layer.
② In Section 4.1, the paper does not present the subscript km of RSSInkm, and does not present the subscript new and superscript k of RSSIn,km,new in Eq. (3).
③ In lines 247-248, there is no explanation of how a 1 × n vector is converted to a 24 × 24 matrix when multiplied by 255.
④ In line 268, the paper does not describe the origin of DS-CNN.
The author's answer: We have improved the expression in section 3.2, corrected the formula in section 4.1, and explained how the 1 × n vector is converted to a 24 × 24 matrix when multiplied by 255 in lines 247-248. When describing DS-CNN, we added more detailed explanations.
- Introduction: The usage of appropriate pictures in the paper can help readers better understand the content of the paper, but the pictures must be clear enough. Figures 1 and Figure 2 in the paper have very low clarity and it is difficult to read the text in them.
- Introduction: There are many errors of detail in the article:
① Equation (1) is inconsistent with the contents.
② The structure of a CNN described in Section 3.2 is inconsistent with Figure 3
③ In line 240, n is not the index of fingerprint collected in the mth acquisition point, n is independent of m.
④ Figure 10 shows real images of rooms 407 and 512, but the two images are identical.
⑤ In Table 2, the last two rows of parameters are clearly duplicated in the first two rows.
⑥ In Figure 13, there is only a loss curve, not an accuracy curve, which is inconsistent with the content.
⑦ In line 372, “Table Ⅳ” should be replaced by “table 4”.
The author's answer: We have corrected all the issues according to the above suggestions. We rewrote the wrong formula in ①. We improved the description of the CNN structure in ②. We solved the errors of expression and format in ③ ④ ⑤ ⑥ ⑦.
- Introduction: There are also many problems with the experimental proof section of the article:
① The results presented in Table 4 are inconsistent with the content.
② The results presented in Table 4 are quite different from those presented in Figure 13 and Figure 14.
The author's answer: The expression in Table 4 has been updated to the correct version, and we have corrected the data errors in the description.
③ Iteration curves for the accuracy and loss values of the proposed method on the UJIIndoorLoc dataset are not shown in the article.
④ In lines 379-384, the author only gives the experimental data, does not present the data in a table, analyze the data in any way, or draw any relevant conclusions.
The author's answer: We apologize for our mistake in editing the paper, which caused the wrong information of the figures. We have corrected it now. Figure 14 shows the accuracy and loss curves of our proposed method on the UJIIndoorLoc dataset. At the same time, we have added tables and data at the changed places, and explained the relevant principles, and summarized the relevant conclusions.
Comments on the Quality of English Language
Introduction: The traces of the usage of translation software in the article are extremely serious, and the author has not meticulously checked the article after translation, resulting in inconsistencies and grammatical errors in many parts of the article:
- In line 178, “expressed” should be replaced by “measured”.
- In line 183, the definite article “the” should be added before "distance" and described "the distance" in detail.
- The terms referred to in lines 285 and 292 should both be “Non-line-of-sight Propagation", but the authors use different terms.
- method column in Table 3, the method proposed in this paper should use “Ours”, not “Our”.
- The authors use passive voice extensively, even in the contributions section, which is bad.
- Authors like to use “this study” as a substitute for themselves, for example in line 159, “this study improves the ...” but the real subject should not be “this study” but “we”. the sentence should be replaced by “we improve the ... in this study.”
- There is improper use of the singular and plural in the article.
The author's answer: We have corrected the suggestions on the English language quality, and re-edited the improper expressions in the language.
Reviewer 4 Report
In this paper, authors propose an indoor localization model that uses a 3D ray-tracing technique to simulate the wireless received signal strength intensity in the field area.
The whole idea is not really new, but the paper introduces some interesting new ideas. Some issues must be fixed in oorder to accept this paper:
- At the end of the conclusion section, authors should include a paragraph explaining the rest of the paper.
- In the related work section I find missing the following reference:
SR Jondhale, V Mohan, BB Sharma, J Lloret, SV Athawale, Support vector regression for mobile target localization in indoor environments, Sensors 22 (1), 358. 2022
- At the end of the related work section, authors should explain what other authors have not done and what is going to be done with this proposal.
- Figure 11, 13, 14, 15 and 16 should be explained in greater detail.
- Authors should include their future work at the end of the conclusion section.
Author Response
Dear reviewer,
We are very grateful for your comments and professional suggestions. These views help to improve the academic rigor of our paper. According to your suggestions and requirements, we have revised and corrected the modified manuscript, and we hope that our work can be improved again. In addition, we would like to explain in detail as follows:
Introduction: At the end of the conclusion section, authors should include a paragraph explaining the rest of the paper.
The author's answer: According to the suggestion, we have improved the rest of our method.
Introduction: In the related work section I find missing the following reference:
SR Jondhale, V Mohan, BB Sharma, J Lloret, SV Athawale, Support vector regression for mobile target localization in indoor environments, Sensors 22 (1), 358. 2022
The author's answer: We have added the relevant references in the corresponding places in the paper.
Introduction: At the end of the related work section, authors should explain what other authors have not done and what is going to be done with this proposal.
The author's answer: According to the suggestion, we have explained the contribution of each author in the related work section.
Introduction: Figure 11, 13, 14, 15 and 16 should be explained in greater detail.
The author's answer: We have further improved the explanation of Figures 11, 13, 14, 15 and 16.
Introduction: Authors should include their future work at the end of the conclusion section.
The author's answer: We have discussed how future work can improve the limitations of the existing method in the end of the paper.
Round 2
Reviewer 1 Report
Thank you for making an effort to address my comments. I believe they have been adequately addressed and the manuscript is now ready for publication.
There are only minor issues with the quality of English in the manuscript.
Reviewer 2 Report
Authors addressed the points raised by the reviewers in the previous round. Now the manuscript of this version has been improved.
The expression is basically understandable.